# Epigenomic Features and Potential Functions of K^+^ and Na^+^ Favorable DNA G-Quadruplexes in Rice

**DOI:** 10.3390/ijms23158404

**Published:** 2022-07-29

**Authors:** Yilong Feng, Zhenyu Luo, Ranran Huang, Xueming Yang, Xuejiao Cheng, Wenli Zhang

**Affiliations:** 1State Key Laboratory for Crop Genetics and Germplasm Enhancement, Collaborative Innovation Center for Modern Crop Production Co-Sponsored by Province and Ministry (CIC-MCP), Nanjing Agricultural University, No.1 Weigang, Nanjing 210095, China; 2017201057@njau.edu.cn (Y.F.); 2019101150@njau.edu.cn (Z.L.); 2021101068@stu.njau.edu.cn (R.H.); xuejiaocheng@njau.edu.cn (X.C.); 2Jiangsu Academy of Agricultural Sciences, Nanjing 210014, China; xmyang@jaas.ac.cn

**Keywords:** BG4-DNA-IP-seq, K^+^-/Na^+^-specific IP-G4, epigenetic features, functional relevance

## Abstract

DNA G-quadruplexes (G4s) are non-canonical four-stranded DNA structures involved in various biological processes in eukaryotes. Molecularly crowded solutions and monovalent cations have been reported to stabilize in vitro and in vivo G4 formation. However, how K^+^ and Na^+^ affect G4 formation genome-wide is still unclear in plants. Here, we conducted BG4-DNA-IP-seq, DNA immunoprecipitation with anti-BG4 antibody coupled with sequencing, under K^+^ and Na^+^ + PEG conditions in vitro. We found that K^+^-specific IP-G4s had a longer peak size, more GC and PQS content, and distinct AT and GC skews compared to Na^+^-specific IP-G4s. Moreover, K^+^- and Na^+^-specific IP-G4s exhibited differential subgenomic enrichment and distinct putative functional motifs for the binding of certain trans-factors. More importantly, we found that K^+^-specific IP-G4s were more associated with active marks, such as active histone marks, and low DNA methylation levels, as compared to Na^+^-specific IP-G4s; thus, K^+^-specific IP-G4s in combination with active chromatin features facilitate the expression of overlapping genes. In addition, K^+^- and Na^+^-specific IP-G4 overlapping genes exhibited differential GO (gene ontology) terms, suggesting they may have distinct biological relevance in rice. Thus, our study, for the first time, explores the effects of K^+^ and Na^+^ on global G4 formation in vitro, thereby providing valuable resources for functional G4 studies in rice. It will provide certain G4 loci for the biotechnological engineering of rice in the future.

## 1. Introduction

DNA G-quadruplex (G4) is preferentially formed at guanine (G)-rich DNA sequences. It is stabilized by Hoogsteen-type hydrogen bonds that are formed among intra- or inter-strands of G-quartets [1]. X-ray diffraction of guanylic acid (GMP) gels, for the first time, led to the prediction of co-planar G-quartets in 1962 [2]. DNA G4 in vitro was discovered when using DNA oligonucleotides corresponding to sequences from immunoglobulin switching regions [3] and telomeres [4]. It is one of the most studied non-B DNA structures in eukaryotic genomes, especially in humans. G4s have been found to preferentially form in various functional genomic regions [5,6], including telomeres [3,7,8,9,10,11], rDNA loci [12,13], promoters [6,14,15], and the first introns of human genes [16].

It has been reported that DNA G4s play regulatory roles in modulating the expression of certain genes, thereby being involved in various biological processes [15,17,18,19,20,21,22]. They include, but are not limited to, DNA replication [10,23], immunoglobulin switch recombination [24], telomere maintenance [10,11], genome instability [10,25], gene transcription [6,15,26,27,28], reprogramming of DNA or chromatin modifications [29,30], human stem cell differentiation [31], and human diseases [32,33]. In particular, they can be considered as potential drug targets for human disease treatment [11,25,34,35], which attracts more attention and boosts G4 studies in pharmacogenomics.

DNA or RNA G4s have been extensively studied in humans using both computational prediction and experimental investigations in vivo and in vitro. Experimental studies include G4-binding antibody-related ChIP-seq (chromatin immunoprecipitation coupled with sequencing) or -chip, such as anti-BG4 [6], hf2 [36], and D1 antibodies [8]; BG4-antibody-based cleavage under targets and tagmentation (CUT&Tag) [37]; G4 ligands, such as template-assembled synthetic G-quartet (TASQ)-related RNA-G4RP (G4-RNA–specific precipitation-sequencing) [38], the binding of an artificially truncated G4 helicase G4-binding domain of the RHAU (DHX36) to G4s [39], and DNA probes-based G4-seq (a G-quadruplex sequencing method) [40]. The presence of putative G-quadruplex sequences (PQSs) in DNAs or RNAs mainly relies on computational screening in most plant species [20,41,42,43], including *Arabidopsis* [44], and grass species, such as maize [45], rice [46], wheat [47], and barley [48]. In addition, very limited studies of plant G4s showed that G4s may play potential roles in regulating various physiological processes in plants, such as gene expression and translation, and in response to various stresses [15,20,28,47,49,50,51]. However, the global identification of G4s in vitro has only been conducted using G4-seq in *Arabidopsis* [40,52], and BG4-DNA-IP-seq (DNA immunoprecipitation with anti-BG4 antibody coupled with sequencing) in rice [15]. Thus, the advance of experimental G4 studies in plants currently lags way behind compared to that in humans.

It has been documented that molecularly crowded solutions, such as polyethylene glycol (PEG), which can mimic the cellular environment [53,54], and monovalent cations, such as K^+^ and Na^+^ [55,56,57], help to stabilize the G4 structure in vivo and in vitro. Moreover, the G4-stabilizing capacity is cation or chemical-dependent. PEG 200 itself has been found to facilitate the G3(T2AG)3 DNA sequence to form antiparallel G4s in vivo [58]. K^+^ is better for G4 formation and stability than Na^+^ [59]. It is hard to predict the ion cation-dependent formation of G4s in vivo or in vitro. In particular, genome-wide profiling of K^+^- and Na^+^-favorable G4 formation is still completely uncharacterized in plants. Therefore, more efforts need to be invested to experimentally validate the folding potential of these PQSs in vitro or in vivo in plants.

In this study, we conducted BG4-DNA-IP-seq following our published procedures [15] for the global identification of in vitro G4 (termed as IP-G4s, representing G4s pulled down by anti-BG4 antibody) under K^+^/Na^+^+PEG 200 conditions. We then compared the genomic distributions, and the genomic and epigenomic features of K^+^- and Na^+^-specific IP-G4s. We finally investigated the possible functional relevance of genes with K^+^- and Na^+^-specific IP-G4s. Thus, our study provides new insights into the ion cation-dependent formation of G4s and their relevant biological functions in rice.

## 2. Results

### 2.1. Identification of G-Quadruplexes under K^+^/Na^+^+PEG200 Conditions

It has been reported that the presence of K^+^ or Na^+^ favors G4 formation in vitro [55,56,57]. However, the differential formation of G4s genome-wide between K^+^ and Na^+^ is still largely unknown in plants. To address this, we conducted BG4-DNA-IP-seq following our published procedures [15] to identify the in vitro formation of G4s under K^+^/Na^+^+PEG200. We sequenced two well-correlated biological replicates (r = 0.94, 0.91, and 0.95 for input, K^+^/Na^+^+PEG, respectively; “r” represents correlation coefficient) for input and K^+^/Na^+^+PEG200 (Appendix A and Appendix A). After peak calling using the same parameters, we found that the overlapping ratio between IP-G4 peaks, representing G4s pulled down by the BG4 antibody, identified from biological replicates was 81.6% for K^+^+PEG200 and 80.4% for Na^+^+PEG200 (Appendix A). We decided to use biologically reproducible IP-G4 peaks (44,845 for K^+^+PEG200, and 37,626 for Na^+^+PEG200) for identifying K^+^- and Na^+^-favorable IP-G4s. We detected three subtypes of IP-G4s: 29,122 common ones, 16,725 K^+^-specific, and 8784 Na^+^-specific ones (Figure 1A). A representative Integrative Genomics Viewer (IGV) illustrates reproducible common and K^+^/Na^+^-specific IP-G4s between replicates (Figure 1B). To assess the accuracy of IP-G4s, we examined ten G4-forming sequences already experimentally validated in rice before [28], and found that seven loci in our study matched well with the amplitude of CD peak in the previous report, including four common IP-G4s, and two K^+^- or Na^+^-specific IP-G4s (Appendix A). Moreover, two common IP-G4s and one Na^+^/K^+^-specific IP-G4 (Appendix A) were further confirmed using circular dichroism (CD) spectroscopy (Figure 1C). Unlike the K^+^-specific or the common IP-G4s, the Na^+^-specific G4 sequence exhibited a similar CD amplitude between Na^+^ and K^+^ conditions. This indicates that Na^+^ and K^+^ had a similar G4 stabilization potential for the Na^+^-specific G4 sequence, which is in contrast to the general trend of K^+^ > Na^+^ in the ability for the formation of G4 in vitro, reflecting that Na^+^ has more ability to facilitate G4 formation from the Na^+^-specific G4 sequence, as compared to K^+^-specific or common IP-G4s. Moreover, the inconsistency between the CD result and the BG4 DNA IP-seq result is possibly caused by the differences of BG4 binding efficiency between G4 formed from the Na^+^-specific G4 sequence under PEG 200/K^+^ and PEG 200/Na^+^ conditions. The common IP-G4s and one Na^+^-specific loci were also confirmed using an IP-G4 dot blotting assay (Appendix A). For example, the immunosignal of Na^+^-specific IP-G4 formed under 150 mM NaCl+PEG200 was stronger than that formed under 150 mM KCl+PEG200 (Appendix A), whereas the CD-spectroscopy assay exhibited no difference between them, which is the same as what has been reported before [28].

### 2.2. Sequence Features and Genomic Distributions of Each Subtype of IP-G4s

To assess if there are any differences in the intrinsic genomic features between Na^+^- and K^+^-specific IP-G4s, we calculated the peak size of three subtypes of IP-G4s mentioned above, and found that the average peak length ranked in a descending order as follows: common IP-G4s (~270 bp), and K^+^- (~250 bp) and Na^+^- (~220 bp) specific IP-G4s (Appendix A). This result showed that IP-G4 peak length variations were ion cation-dependent. We then calculated GC content, and GC and AT skews around ±1 kb from the start to the end point of the IP-G4 regions. We found that the mean level of GC content in a descending order was common IP-G4s, and K^+^ and Na^+^-specific IP-G4s (Figure 2A). Regarding GC and AT skews, we observed that three subtypes of IP-G4s exhibited a similar profile of AT skew, but had a distinct profile of GC skew. K^+^-specific IP-G4s had a clear GC skew distributed at the start or end of the IP-G4 regions; by contrast, the GC skew of Na^+^-specific IP-G4s was shifted to approximately ±200 bp from the start or the end point (Figure 2B).

To investigate the genomic distributions of each subtype of IP-G4s, we partitioned the whole genome into seven functionally annotated sub-genomic regions, such as promoters, 5′UTRs, exons, introns, downstream of the transcriptional end sites (TESs), and intergenic regions. Distinct subgenomic distributions occurred among three subtypes of IP-G4s (Figure 2C). For instance, common IP-G4s had the highest percentage in promoters and 5′ untranslated regions (UTRs); Na^+^-specific IP-G4s had the highest percentage in distal intergenic regions and downstream of transcriptional start sites (TTSs); and K^+^-specific IP-G4s had the highest percentage in exons and introns. After calculating the fold enrichment between the observed and expected ones (bedtools shuffle command, observed values divided by the average of three randomization values), we found that all subtypes of IP-G4s were significantly overrepresented in promoters and 5′UTRs; by contrast, only K^+^-specific IP-G4s exhibited a little enrichment in intergenic and downstream of TTSs regions (Figure 2D). Thus, these results indicate that K^+^ and Na^+^ favorable IP-G4s have distinct sequence features and subgenomic enrichment in rice.

### 2.3. Distinct PQS Features among Subtypes of IP-G4s

It has been documented that there are canonical and non-canonical types of PQSs in eukaryotic genomes. They have structural variations strictly depending on their underlying DNA sequences, such as the length and the constituent bases of the loops, base modifications, the strand direction, and the number of G tracks [60]. To assess if there are any differences of PQS subtypes in each subtype of IP-G4s, we identified PQSs within IP-G4 peaks using the regular expression, as previously reported [41]. Each subtype of IP-G4s had more G2L1-7 (PQSs with two tetrads containing a loop length between 1 and 7 nucleotides), but less G3L8-12 (extended canonical PQSs containing at least one loop with a length between 8 and 12 nucleotides) (Appendix A). After comparing the fold enrichment over random controls, we found that G3L8-12 had the highest fold enrichment in each subtype of IP-G4s (Figure 3A). In general, K^+^-specific IP-G4s had a higher percentage of each subtype of PQSs than Na^+^-specific ones, especially for G3L1-7 (canonical PQSs with three tetrads containing a loop length between 1 and 7 nucleotides) (Figure 3A). In terms of the percentage of each subtype of PQSs, a subtle difference was observed among three subtypes of IP-G4s (Figure 3B). For example, common IP-G4s had a slightly higher percentage of G2L8-12 (PQSs with two tetrads including at least one loop with a length between 8 and 12 nucleotides) and G3L1-7 than the other two. Na^+^-specific IP-G4s had 1.4% more non-canonical types of PQSs than K^+^-specific ones.

After examining the subgenomic distributions of each subtype of PQSs, we found that K^+^-specific IP-G4s had the highest percentage of G3L1-7 and G3L8-12 distributed in intergenic regions and downstream of TTSs, whereas common IP-G4s had the highest percentage of G3L1-7 and G3L8-12 distributed in promoters and 5′UTRs (Figure 3C). Na^+^-specific IP-G4s had more G3L1-12 (extended canonical PQSs with three tetrads and a loop length between 1 and 12 nucleotides) in introns and promoters than K^+^-specific ones (Figure 3C). A similar trend regarding the subgenomic distributions of each subtype of PQSs among three subtypes of G4s also occurred for G2L1-7 and G2L8-12 (Appendix A).

G4s were reported to potentially act as cis-regulatory elements [6,61,62,63], and serve as transcription factor binding hubs in humans [64]. To examine if K^+^- and Na^+^-specific IP-G4s harbor any distinct motifs for the recruitment of trans-acting factors, we conducted a de novo motif identification by screening K^+^- and Na^+^-specific IP-G4-related sequences using MEME-ChIP [65]. We detected that GAGA and GGC motifs corresponding to the binding of BPC1 and ERF5 were significantly enriched in K^+^-specific IP-G4s; by contrast, GGG and AAA motifs corresponding to the binding of lateral organ boundaries domain (LOB) and vernalization1 (VRN1) were significantly enriched in Na^+^-specific IP-G4s (Figure 3D). Differential motif enrichment between K^+^- and Na^+^-specific IP-G4s indicates that K^+^- and Na^+^-specific IP-G4s may have distinct biological functions in rice.

Collectively, these results indicate that K^+^ and Na^+^ favorable IP-G4s have distinct PQS content, and variations in subgenomic enrichment occur for each subtype of PQSs in rice.

### 2.4. Distinct Epigenomic Features between K^+^- and Na^+^-Specific IP-G4s 

Our previous study showed that G4s have some unique epigenomic features [15]. However, the epigenomic features of K^+^- or Na^+^-specific IP-G4s are still unclear. To this end, we calculated CG, CHG, and CHH methylation levels at ±1 kb of the midpoint of each subtype of IP-G4 with similar C content, as indicated in Figure 4A. We found that Na^+^-specific IP-G4s exhibited higher CG and CHG, but less CHH methylation levels, across all regions examined than K^+^-specific IP-G4s (Figure 4A). We found that Na^+^-specific IP-G4s had 3.2% less overlapping DNaseⅠhypersensitive sites (DHSs), but 5.0% more overlapping R-loops than K^+^-specific IP-G4s (Appendix A). We then performed similar analyses for DNA-6mA, DHSs, and R-loops as we did for DNA methylation. We found that Na^+^- and K^+^-specific IP-G4s were less enriched with DNA-6mA relative to the random control, and even less for K^+^-specific ones (Figure 4B, top); Na^+^-specific IP-G4s had higher DH abundance at the center, but had less DH abundance at the immediate flanking regions than K^+^-specific IP-G4s (Figure 4B, middle); Na^+^-specific IP-G4s had higher R-loop abundance at center than K^+^-specific IP-G4s (Figure 4B, bottom).

We then investigated 13 histone marks and the nucleosome occupancy for Na^+^/K^+^-specific IP-G4s at ±1 kb of the center of each subtype of IP-G4. We found that K^+^-specific IP-G4s had more enrichment levels of H3K23ac, H3K4ac, H3K9ac, H4K16ac, and H3K4me3, and had similar H3K27ac enrichment, but had less enrichment levels of H3K4me2, H4K12ac, and H3K9me1/2/3, as compared to Na^+^-specific IP-G4s (Figure 4C). Moreover, Na^+^/K^+^-specific IP-G4s were less enriched with H3K27me3 and H3K36me3, and had lower nucleosome occupancy relative to the random control, and even less for K^+^-specific ones (Figure 4C). Furthermore, we conducted fold changes for all of the marks mentioned above for Na^+/^K^+^-specific IP-G4s distributed at promoters and gene bodies. We found that K^+^-specific IP-G4s had slightly higher enrichment levels of H3K4/9/23/27ac, H3K4me3, and H4K16ac, but had less enrichment levels of H3K9me1/3 at promoters and gene bodies, as compared to Na^+^-specific IP-G4s (Figure 4D). In addition, we associated different types of reported TADs (topologically associated domains) [66] with each subtype of IP-G4s, and found that Na^+^-specific IP-G4s had a higher percentage of active domains, but a slightly lower percentage of repressive domains, as compared to K^+^-specific IP-G4s (Appendix A).

Taken together, these results show that K^+^- and Na^+^-specific IP-G4s have distinct epigenomic features: the former tends to be more associated with active marks than the latter.

### 2.5. Potential Involvement of K^+^- and Na^+^-Specific IP-G4s in the Regulation of Gene Transcription

G4s can play repressive and active roles in the regulation of gene transcription in rice [15]. To further assess if K^+^- and Na^+^-specific IP-G4s have distinct biological functions, we associated K^+^-/Na^+^-specific IP-G4s with TE genes (TEGs) and non-TE genes (non-TEGs), and found that the number of TEGs (4675 vs. 2454) and non-TEGs (1676 vs. 1398) overlapping K^+^-specific IP-G4s was higher relative to the ones overlapping Na^+^-specific IP-G4s (Appendix A). Among the TEGs analyzed, 35.5% of Ty were associated with K^+^-specific IP-G4s, which was 4.7% more than Na^+^-specific IP-G4s (Appendix A). Moreover, we examined the subtypes of TEs overlapping each subtype of IP-G4s, and found that 14.5% of DNA TEs and 51.7% of LTRs (long-terminal repeats) were overlapping K^+^-specific IP-G4s, which is 4.9% and 9.1% more than the corresponding ones overlapping Na^+^-specific IP-G4s, respectively (Appendix A).

To examine the impacts of each subtype of IP-G4s on the expression of overlapping non-TEGs, we divided all TEGs and non-TEGs into expressed (FPKM (fragments per kilobase million mapped fragments) > 0) and non-expressed (FPKM = 0) ones according to the FKMP values. We found that K^+^-specific IP-G4s were overlapping more expressed non-TEGs (71.8% vs. 67.4%) and less non-expressed (28.2% vs. 32.7%) non-TEGs than Na^+^-specific IP-G4s (Appendix A); and that the expressed and non-expressed TEGs overlapping K^+^-specific IP-G4s were similar to the ones overlapping Na^+^-specific IP-G4s (Appendix A). These results indicated that K^+^-specific IP-G4s tended to play active roles in the regulation of non-TEGs. To further confirm this, we compared the expression levels of non-TEGs and TEGs overlapping each subtype of IP-G4s, and found that the mean expression levels of genes overlapping K^+^-specific IP-G4s tended to be higher than those overlapping Na^+^-specific IP-G4s (Figure 5A). Moreover, the mean expression levels of expressed genes overlapping common IP-G4s at promoters or gene bodies were higher than the corresponding ones overlapping Na^+^-specific IP-G4s (Figure 5B). We then plotted normalized IP-G4 read counts across ±1 kb from the TSS to the TES of K^+^- or Na^+^-specific IP-G4 overlapping genes with different expression levels (high, medium, low, and non-expressed; FPKM values). We found that K^+^-specific IP-G4s at the promoter or gene body exhibited a positive or negative correlation with the expression levels of overlapping genes, respectively (Figure 5C, left), which is similar to the previous findings [15], whereas this was not the case for the association between Na^+^-specific IP-G4 at the promoter or gene body with the expression levels of overlapping genes (Figure 5C, right). For instance, the IP-G4 read intensity was similar between genes with high and low expression levels at the promoter, and between genes with high and middle expression levels at gene bodies.

We next extracted genes with promoters or gene bodies overlapping K^+^- or Na^+^-specific IP-G4s (Appendix A) for GO (gene ontology) term enrichment analyses. We found that genes with promoters overlapping K^+^-specific IP-G4s had more enriched GO terms associated with biological processes, cellular components, and molecular functions when compared with genes with promoters overlapping Na^+^-specific IP-G4s (Figure 5D, left). Moreover, distinct GO term enrichment occurred between genes with the gene body overlapping K^+^- and Na^+^-specific IP-G4s. For example, genes with the gene body overlapping K^+^-specific IP-G4s had enriched GO terms associated with transcription factor/regulator activity, the regulation of gene expression, biological processes, and cell/cell part; by contrast, genes with the gene body overlapping Na^+^-specific IP-G4s had overrepresented functions in RNA/nucleic acid/chromatin binding, DNA metabolic processes, and biosynthetic processes (Figure 5D, right).

Taken together, these results suggest that K^+^- and Na^+^-specific IP-G4s may have distinct associations with the expression of overlapping genes, and distinct biological functions in rice.

## 3. Discussion

The presence of PQSs in the genome does not truly represent G4 formation in vitro or in vivo under different favorable conditions. Therefore, it is necessary to provide experimental evidence showing the G4 folding potential in vivo or in vitro, especially on a genome-wide scale. Our study showed that K^+^+PEG was more favorable for in vitro G4 folding than Na^+^+PEG (Figure 1A and Appendix A), and Na^+^ facilitates more G4s formed from non-canonical PQSs than K^+^. It has been reported that K^+^ is better to facilitate G4 formation and stability than Na^+^ [59]; K^+^ is preferable for parallel G4 formation, whereas Na^+^ facilitates antiparallel G4 folding [28]. The combination of K^+^ and PEG was also found to be favorable for the formation and maintenance of G4s in vivo [54,67]. Plausible explanations for this are as follows: one is that K^+^ is more favorable to interact with O6 atoms of guanine, and needs a lower dehydration energy to form Hoogsteen base pairing to connect adjacent guanosines when compared with Na^+^ [68]. Another possible explanation is that it is caused by the distinct intrinsic underlying DNA sequences capable of folding G4s. Our study showed that K^+^-specific IP-G4s had more GC content, a longer peak size, more PQSs density, and a distinct GC skew, as compared to Na^+^-specific IP-G4s (Figure 2 and Appendix A). Thus, G4 formation in vitro can be determined by the intrinsic DNA sequence features alone or combined with ionic types. A similar phenomenon could be applied for G4 formation in vivo. Plant cells usually maintain K^+^ and Na^+^ homeostasis for ensuring normal growth and development. Na^+^ and K^+^ have been documented to have contrasting effects on cell functions, and plant growth and development; thus, the maintenance of cellular Na^+^ and K^+^ homeostasis is essential for normal cell function and plant development [69,70]. The unbalance of cytosolic Na^+^ and K^+^ concentration occurs under stress conditions [71,72,73]. Thus, the presence of Na^+^- and K^+^-specific IP-G4s in vitro infers that the disruption of Na^+^ and K^+^ homeostasis under abnormal physiological conditions, such as biotic and abiotic stresses, may induce the dynamics of Na^+^- and K^+^-specific G4 formation in vivo, thereby resulting in changes in some G4-related biological events. Thus, our results, for the first time, provided evidence at a genome-wide level showing the distinct impacts of K^+^ and Na^+^ on the folding of G4s in vitro in rice. Our study simulated K^+^- and Na^+^-specific conditions for the in vitro formation of G4s. It is worth noting that it cannot truly represent the effects of K^+^ and Na^+^ on G4 formation in vivo.

The regulatory roles of G4s in gene transcription have been reported in multiple species, including humans [6,74,75,76,77] and *E. coli* [78], and are speculated to exist in maize [45,79], rice [15], and *Arabidopsis* [20,49]. Moreover, genes containing PQSs are possibly involved in stress conditions, such as hypoxia, low sugar and nutrient deprivation [45], and several signaling pathways [20]. However, little is known about the potentially distinct functions of genes with K^+^- and Na^+^-specific G4s in plants. Our study showed that genes with K^+^-specific IP-G4s were more expressed than those with Na^+^-specific IP-G4s, suggesting that K^+^-specific IP-G4s tend to activate the expression of overlapping genes. The roles of K^+^- or Na^+^-specific IP-G4s in the regulation of gene transcription are possibly mediated by the following factors: differential enrichment of motifs for TF binding, suggesting that both subtypes of G4s could serve as a platform for the recruitment of distinctly functional trans-actors; differential enrichment of various epigenomic marks (Figure 4). In general, K^+^-specific IP-G4s were more associated with active marks, such as active histone marks, and low DNA methylation levels, as compared to Na^+^-specific IP-G4s; thus, K^+^-specific IP-G4s, in combination with active chromatin features, facilitate the expression of overlapping genes. It has been reported that G4s are co-localized with open chromatin [6] and R-loops [29,80]. In addition, K^+^- and Na^+^-specific IP-G4 overlapping genes exhibited distinct GO term enrichment (Figure 5D). Thus, each subtype of IP-G4s may coordinate with epigenomic marks to modulate the transcription of overlapping genes, thereby exhibiting distinct biological functions.

Together, our study will pave the way for promoting the functional characterization of G4s in rice, and for potential applications of certain G4 loci for the biotechnological engineering of rice in the future.

## 4. Materials and Methods

### 4.1. Rice Seedlings

The rice (*Oryza. sativa* L. spp. japonica) seeds were pre-germinated at room temperature (RT). Uniformly germinated seeds were transferred into soil for growing in a greenhouse at a temperature of 28–30 °C and a 14 h/10 h light/dark cycle. Two-week-old rice seedlings were collected and cut into small slices of 1–1.5 cm in size, followed by cross-linking with a 1% final concentration of formaldehyde for 10 min under vacuum. The cross-linked samples were ground into fine powder in liquid nitrogen. Ground powder can be stored in −80 °C for use later or immediately used for genomic DNA preparation.

### 4.2. Dot Blotting Assays

The synthesized DNA oligoes corresponding to common and Na^+^/K^+^-specific IP-G4s (Appendix A) were denatured at 95 °C for 5 min in the G4-stabilizing buffer (40% PEG 200 and 10 mM Tris-HCl, pH = 7.5, with 150 mM KCl or NaCl), as indicated, and were then gradually cooled down to RT for reassociation. The reassociated DNA oligoes were loaded onto the Amersham Hybond-N^+^-nylon membrane and preblocked in 5% milk for 45min at RT. The preblocked membrane was incubated with anti-BG4-FLAG antibody overnight at 4 °C, followed by incubation with anti-FLAG antibody for an additional 1.5 h. After washing three times, the membrane was stained with horseradish peroxidase (HRP)-conjugated anti-rabbit secondary antibody (1:10,000), followed by immuno-signal development and recording.

### 4.3. Circular Dichroism (CD)-Spectroscopy

The DNA oligoes corresponding to common and Na^+^/K^+^-specific IP-G4s (Appendix A) were denatured at 95 °C for 10 min in K^+^/Na^+^ buffer (pure water with 150 mM KCl or NaCl), as indicated, and were then gradually cooled down to RT for reassociation overnight. For CD-spectroscopy, 5 μM oligoes in pure water containing 150 mM KCl/NaCl were scanned at a wavelength range of 220–320 nm, with 1 nm bandwidth, a response time of 0.25 s, and a path length of 1 mm on a Chirascan Spectropolarimeter (Applied Photophysics, Hachioji, Tokyo, Japan). Data were buffer-subtracted, normalized to provide molar residue ellipticity values, and smoothed.

### 4.4. BG4-DNA-IP-Seq

BG4-DNA-IP-seq, DNA immunoprecipitation with anti-BG4 antibody coupled with sequencing, was conducted following our previously published protocols [15]. Briefly, a total of 5 μg of fragmented genomic DNA was denatured and reassociated in the G4 stabilization buffer (40% PEG200, 10 mM Tris-HCl, pH = 7.5, with 150 mM KCl or NaCl). The reassociated DNA-G4s were specifically recognized by using anti-BG4-FLAG antibody in the IP incubation buffer (50 mM HEPES, 150 mM KCl, 1 mM MgCl_2_, 130 nM CaCl_2_, 1% BSA, 40% PEG200, Complete Mini, pH = 7.5), which was followed by incubation with an additional anti-FLAG antibody (D110005, BBI, Shanghai, China) and washed protein G Dynalbeads (10004D, Invitrogen, Carlsbad, CA, USA). Protein-G-bead-bound anti-BG4-DNA-G4s complexes were finally collected and washed three times with washing buffer (10 mM Tris-HCl, 150 mM KCl, 1% Tween20). The BG4-bound DNA-G4s were finally eluted using 200 μL elution buffer (0.1 M NaHCO_3_, 1% SDS) two times. Each experiment was biologically repeated. The BG4-DNA-IPed DNA or input DNA was recovered for library preparation. All libraries were prepared using the NEBNext^®^ Ultra™ II DNA Library Prep Kit for Illumina (E7645S, NEB, Lpswich, USA), and sequenced using the Illumina platform, followed by data analyses.

### 4.5. Analyses of BG4-DNA-IP-Seq Data 

Raw data were cleaned using Trim Galore! (Version 0.4.4, Felix Krueger, Cambridge, UK) for the removal of low-quality reads and adapter contamination. BWA (Burrows-Wheeler Aligner) (mem algorithm, version 0.7.17, Li and Durbin, Cambridge, UK) with default parameters was used to align all cleaned reads to the MSU v7.0 reference genome (http://rice.plantbiology.msu.edu/pub/data/Eukaryotic_Projects/o_sativa/annotation_dbs/pseudomolecules/version_7.0/all.dir/ (accessed on 20 June 2022)). SAMtools (version 1.5, Li, Heng, et al, Cambridge, UK, option -markdup) was used to remove PCR duplicates. MACS (version 2.1.1, Zhang Y, et al, Bosto, MA, USA) was used to call IP-G4 (referred to as G4s pulled down by anti-BG4 antibody) peaks from reads with an alignment length greater than 50. The input library was used as control, and the command and parameters used were: macs2 callpeak-g 3.8 × 10^8^-f BAM-extsize -p 1×10^−3^-nomodel. Biologically replicated G4 peaks were considered IP-G4 peaks with high confidence (command: intersect of the bedtools package). The plotCorrelation program of deepTools was used to calculate the Spearman’s rank correlation coefficients between biological replicates in input and K^+^/Na^+^+PEG200 conditions.

### 4.6. Analyses of Public Omics Data Sets

**ChIP-seq:** 13 histone marks were reanalyzed following our previously published procedures [81]. **DNase-seq:** DNase-seq data were reanalyzed using the procedures previously described [82]. **DRIP-seq and BS-seq:** both data were reanalyzed according to the published protocols [81]. C/(C+T) and the methylation levels were calculated using similar procedures as previously described [81]. **DNA-6mA IP-seq:** 6mA IP-seq data were reanalyzed following the published protocols [83]. **MNase-seq**: MNase-seq data were reanalyzed according to the published protocols [84]. All public data used in this study are listed in Appendix A.

### 4.7. Motif Prediction

IP-G4-related motifs located at ±100 bp around the center of IP-G4 peaks were identified using MEME-ChIP (http://meme-suite.org/tools/meme-chip (accessed on 26 June 2022)) [65], with the following parameters: minimum width, 5; and maximum width, 15. The identified motifs were used to screen the *Arabidopsis* database for the identification of putative TF-binding sites (Tomtom tool). Only the top two significantly enriched motifs with the highest *E*-values are listed in the text.

### 4.8. PQSs Identification and Fold Enrichment Analyses

Putative G-quadruplex sequences (PQSs) were identified by screening the whole genome sequences using fastaRegexFinder.py (https://github.com/dariober/bioinformaticscafe/blob/master/fastaRegexFinder.py (accessed on 27 June 2022)) [85]. PQSs were classified into subtypes according to the loop length and G repeats. The fold enrichment levels of each subtype of PQSs were calculated by comparing with the random control across the genome (bedtools shuffle command, observed values divided by the average of 100 randomization values).

### 4.9. Analyses of GC Content and GC/AT Skews

GC content (GC %) and GC/AT skews were calculated using the following formulas: GC skew = (G − C)/(G + C), AT skew = (A − T)/(A + T), GC content = (C + G)/(A + T + C + G); the IP-G4 peak regions were equally divided into 50 bins, and the ±1 kb around IP-G4 peaks were divided into 20 bp windows. Random regions were chosen from any genomic regions without IP-G4s (command: shuffleBed in bedtools; option: -noOverlapping).

## Figures and Tables

**Figure 1 ijms-23-08404-f001:**
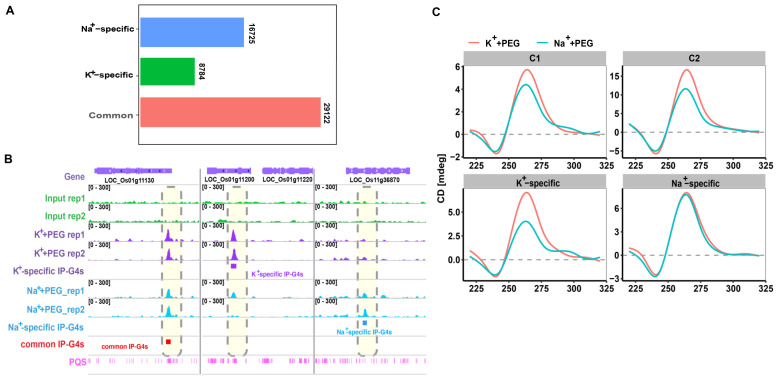
Identification of common and K^+^-/Na^+^-specific IP-G4s. (**A**) Histogram illustrating the number of three subtypes of IP-G4s (common and K^+^-/Na^+^-specific IP-G4s). (**B**) A representative Integrative Genomics Viewer (IGV) snapshot spanning a 21 kb window, illustrating distributions of common and K^+^-/Na^+^-specific IP-G4 peaks in the rice genome. Each significant IP-G4 peak is marked with a colored rectangular box below the peak. Putative G-quadruplexes sequences (POSs) were computationally predicted in the rice genome. (**C**) Circular dichroism (CD) spectroscopy assays using synthesized oligoes for common and K^+^-/Na^+^-specific IP-G4s, respectively.

**Figure 2 ijms-23-08404-f002:**
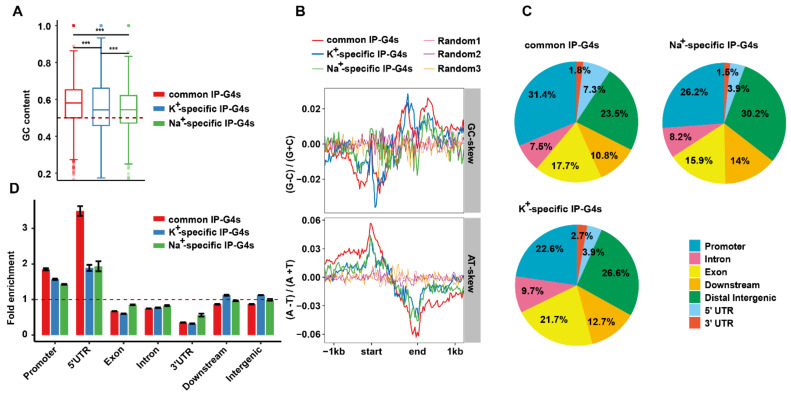
Genomic distributions and sequence features of K^+^- and Na^+^-specific and common IP-G4s. (**A**,**B**) GC content (**A**), and GC and AT skews (**B**) of K^+^- and Na^+^-specific and common IP-G4s. GC content, and GC and AT skews in the forward and reverse DNA strand were calculated at around ±1 kb from three subtypes of IP-G4 peaks. The random peaks were shuffled from IP-G4 peaks in the rice genome. (**C**) Distributions of IP-G4s in different sub-genomic units in the rice genome, including promoters, 5′UTRs, exons, introns, 3′UTRs, downstream of the TESs, and intergenic regions. (**D**) Distributions of the observed IP-G4s over the expected G4s in each sub-genomic unit in the rice genome. The red dash line represents the ratio as 1.0. The significance was determined using the Wilcoxon rank-sum test. *** *p*-value < 0.001.

**Figure 3 ijms-23-08404-f003:**
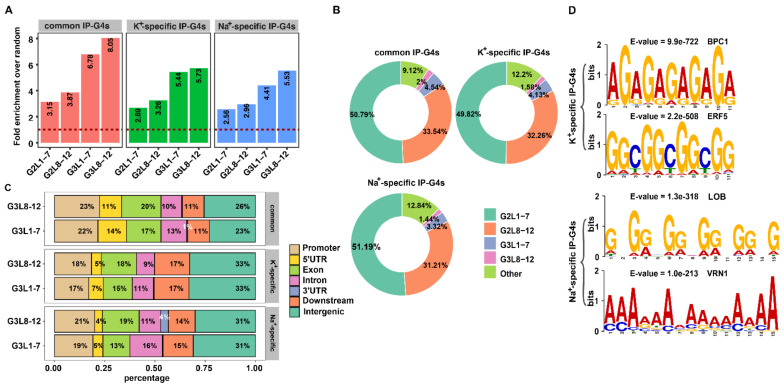
Characterization of PQSs in K^+^- and Na^+^-specific and common IP-G4s. (**A**) Fold enrichment of each subtype of PQSs, as indicated in K^+^- and Na^+^-specific and common IP-G4s. (**B**) The percentage of each subtype of PQSs, as indicated in K^+^- and Na^+^-specific and common IP-G4s. (**C**) Subgenomic distributions of each subtype of G3-related PQSs, as indicated in K^+^- and Na^+^-specific and common IP-G4s. The whole genome was divided into seven functionally annotated regions, including promoters, 5′UTRs, exons, introns, 3′UTRs, downstream of the TESs, and intergenic regions. (**D**) Motif discovery using MEME for K^+^- and Na^+^-specific IP-G4 peaks (random 1000 from the top 5000 peaks ranked by the enrichment levels). The top two significantly enriched motifs are listed, which were identified in IP-G4 regions. E-values are provided at the top.

**Figure 4 ijms-23-08404-f004:**
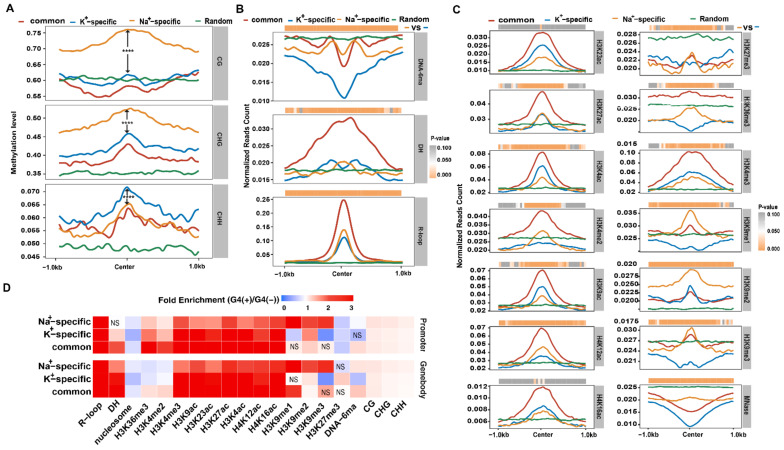
Epigenomic features of K^+^- and Na^+^-specific and common IP-G4s. (**A**) CG, CHG, and CHH methylation levels across ±1 kb of the center of K^+^-specific, Na^+^-specific IP-G4s, and common IP-G4s and random control with similar C content. (**B**) Distributions of normalized read counts of DNA-6mA (6mA), DNase-seq (DH), and DRIP-seq (R-loop) across ±1 kb of the center of K^+^- and Na^+^-specific, common IP-G4s, and random control. The heat map at the top indicates the Wilcoxon rank-sum test between K^+^- and Na^+^-specific IP-G4s, and the color key represents the *p*-values. (**C**) Distributions of normalized read counts of ChIP-seq with 13 histone marks and MNase-seq across ±1 kb of the center of K^+^- and Na^+^-specific IP-G4s, common IP-G4s, and random control. The heat map at the top indicates the Wilcoxon rank-sum test between K^+^- and Na^+^-specific IP-G4s, and the color key represents the *p*-values. (**D**) Heat map showing the enrichment levels of each mark for K^+^- and Na^+^-specific and common IP-G4s at promoters and gene bodies; the color key represents the fold enrichment of each subtype of IP-G4s relative to IP-G4s-. A chi-squared test was conducted to determine the significance of the differences (*p*-value < 0.05). NS represents non-significance. The significance in Figure 4A was determined using the Wilcoxon rank-sum test. **** *p*-value < 0.0001.

**Figure 5 ijms-23-08404-f005:**
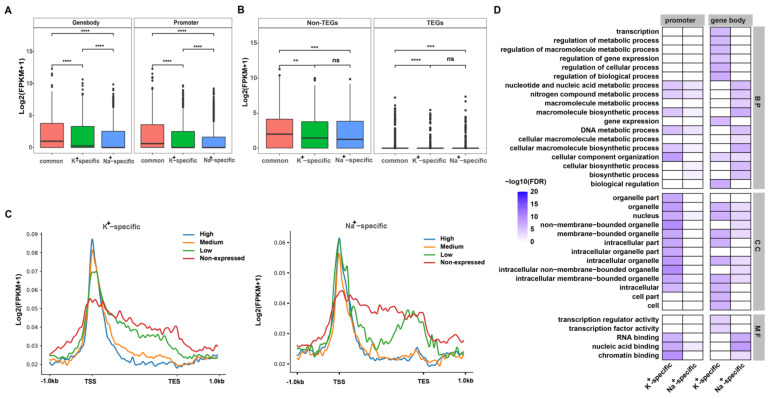
Relationships between each subtype of IP-G4s and transcription of overlapping genes. (**A**) Comparisons of the expression levels of TEGs and non-TEGs overlapping each subtype of IP-G4s. (**B**) Comparisons of the expression levels of genes overlapping each subtype of IP-G4s at promoters and gene bodies. (**C**) Curve plots showing the profile of normalized read counts of K^+^- and Na^+^-specific IP-G4s from 1 kb upstream of TSSs to 1 kb downstream of TESs of overlapping genes with different fragments per milobase per million mapped fragments (FPKM) values (high, medium, low, and non-expressed). (**D**) GO terms enrichment assay using K^+^- and Na^+^-specific IP-G4 overlapping genes. ** *p*-value < 0.01, *** *p*-value < 0.001, **** *p*-value < 0.0001.

## Data Availability

The data generated in this study have been submitted to the NCBI Gene Expression Omnibus (GEO; http://www.ncbi.nlm.nih.gov/geo/) under accession number GSE209990.

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
