# Peer review of "Epigenomic Features and Potential Functions of K+ and Na+ Favorable DNA G-Quadruplexes in Rice"

_ijms, 2022, doi:10.3390/ijms23158404_

Round 1

Reviewer 1 Report

Feng et al. wrote an article about a genome-wide study of G4 sequences regarding their preference for K+ or Na+ conditions. Their research uses several sources of evidence to show that G4s, with and without ionic condition preferences, differ significantly in terms of genomic loci, sequential composition, etc. The authors did a great job (including statistical testing of hypotheses) in convincing the reader of those facts. The discussion section is also a great addition as it tries to present different views on the possible implications or reasons for these G4s preferences. I thank the authors for such great work, and I do not have any significant comments regarding the manuscript. Only two minor comments:

  • Figure 1C. The CD spectrum of the Na+-specific G4 sequence seems almost identical in K+ and Na+ conditions. Why is that?
  • Figure 1A. I am not too fond of this Venn diagram. The K+ and Na+ sets are different, so you cannot intersect them. The intersection segment on your Figure panel has three values (28120, 29122, 28842) precisely because of this reason.

Author Response

Reviewer 1: Comments and Suggestions for Authors Feng et al. wrote an article about a genome-wide study of G4 sequences regarding their preference for K+ or Na+ conditions. Their research uses several sources of evidence to show that G4s, with and without ionic condition preferences, differ significantly in terms of genomic loci, sequential composition, etc. The authors did a great job (including statistical testing of hypotheses) in convincing the reader of those facts. The discussion section is also a great addition as it tries to present different views on the possible implications or reasons for these G4s preferences. I thank the authors for such great work, and I do not have any significant comments regarding the manuscript. Response: We highly appreciate your positive feedback. Thank you! Only two minor comments: • Figure 1C. The CD spectrum of the Na+-specific G4 sequence seems almost identical in K+ and Na+ conditions. Why is that? Response: We thank the reviewer for pointing this out. The Figure 1C showed that the Na+-specific G4 sequence exhibited a similar CD amplitude between Na+ and K+ conditions. This indicates that Na+ and K+ had a similar G4 stabilization potential for the Na+-specific G4 sequence, which is in contrast to the general trend of K+ > Na+ in the ability for formation of G4 in vitro, reflecting that Na+ has more ability to facilitate G4 formation from the Na+-specific G4 sequence as compared to K+-specific or common IP-G4s. Moreover, inconsistence between the CD result and the BG4 DNA IP-seq result is possibly caused by differences of BG4 binding efficiency between G4 formed from the Na+-specific G4 sequence under PEG 200/K+ and PEG 200/Na+ conditions. We added more explanations in the text. • Figure 1A. I am not too fond of this Venn diagram. The K+ and Na+ sets are different, so you cannot intersect them. The intersection segment on your Figure panel has three values (28120, 29122, 28842) precisely because of this reason. Response: We thank the reviewer for pointing this out. We totally agree with you. We used histogram to show common and K+-/Na+-specific IP-G4s. Thank you!

Reviewer 2 Report

The authors are correct in saying that the high interest on G4 is mainly devoted to humans and that there is a need for studies dedicated to plants. The study done is sound, robust and well documented and discussed.

On the whole, I think that the paper deserves publication.

There is only one weak point in the manuscript presented, that should be addressed before publication.

As it is, the text is a little messy. The authors should very carefully re-read their paper and control that any acronym is clear and has been duly defined the first time used. There are many acronyms all along the manuscript, and this lets the work non so easy to follow for non-experts. Authors should better provide an acronym list at the beginning (or end) of the text.

I will upload a file where some acronyms and typos are highlighted. For instance, lines 48-51 contain many non-defined acronyms, PEG is not defined (61), BG4-DNA-IP-seq is not explained (70), r (84), IGV (92), CD (95)...and so on. Authors cannot count on definitions which are provided a long time afterwards, for instance on part 4.

Author Response

Reviewer 2: Comments and Suggestions for Authors The authors are correct in saying that the high interest on G4 is mainly devoted to humans and that there is a need for studies dedicated to plants. The study done is sound, robust and well documented and discussed. On the whole, I think that the paper deserves publication. Response: We greatly appreciate your affirmation on our work. Thank you! There is only one weak point in the manuscript presented, that should be addressed before publication. As it is, the text is a little messy. The authors should very carefully re-read their paper and control that any acronym is clear and has been duly defined the first time used. There are many acronyms all along the manuscript, and this lets the work non so easy to follow for non-experts. Authors should better provide an acronym list at the beginning (or end) of the text. I will upload a file where some acronyms and typos are highlighted. For instance, lines 48-51 contain many non-defined acronyms, PEG is not defined (61), BG4-DNA-IP-seq is not explained (70), r (84), IGV (92), CD (95)...and so on. Authors cannot count on definitions which are provided a long time afterwards, for instance on part 4. Response: Following your suggestions, we provided full names for all acronyms listed in the manuscript. Thank you!